# Reputation Cues as Signals in the Sharing Economy

**Sonny Rosenthal *** 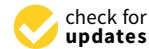**, Jean Yi Colette Tan and Ting Fang Poh**

Wee Kim Wee School of Communication and Information, Nanyang Technological University, Singapore 639798, Singapore; colettan@hotmail.com (J.Y.C.T.); ptingfang@gmail.com (T.F.P.)
* Correspondence: sonnyrosenthal@ntu.edu.sg

**Abstract:** Reputation cues, like star ratings, signal qualities of service providers in the sharing economy and may affect user behavior. Guided by concepts from signaling theory and using a repeated measures experiment (N = 221), this study manipulated the level of star ratings of ride sharing drivers. Intuitive findings are perceived service quality and willingness to use the service provider are higher when the star rating is high versus low. Extending prior work, perceived service quality mediates the effect of reputation on willingness, explaining 83% of the total effect. Also, the direct effect of reputation cues on perceived service quality depends, albeit weakly ($\eta^2_p = 0.02$), on how much users say they pay attention to them. These novel findings clarify the kinds of mental processing that occur when users of shared services evaluate reputation cues. We discuss findings in terms of costly signaling and consider practical implications for users and providers.

**Keywords:** sharing economy; ride sharing; service quality; reputation; signaling theory

## 1. Introduction

Advancements in information and communication technologies have supported growth of the sharing economy. In the sharing economy, individuals provide an underutilized resource—for example, a good or service—for others to use, often in exchange for something of value. Hamari et al. (2016) described this kind of activity as collaborative consumption involving the use of technological platforms to coordinate service providers and users from within a sharing community. Such activity benefits from systems that allow service providers to signal their positive attributes, such as service quality (Ert et al. 2016; Xie and Mao 2017). These kinds of signals are important for users when they have limited information about the provider or service.

In most consumer transactions, sellers have ample information about the goods or services they provide, while buyers have limited knowledge (Cheung et al. 2014). When facing such information asymmetries, buyers often look for information that can help them decide if they wish to engage in the transaction. On some platforms, such as Airbnb, service providers can share detailed personal information, which creates social presence and enhances user trust (Cho et al. 2019). With other services, such as ride sharing, users may need to make quicker judgments based on less information. Reputation cues like star ratings can function as signals on those platforms, often about the quality of service providers (Xie and Mao 2017), helping users make confident decisions.

The current study exandividuals to engage in dmines decision making in the context of the Singaporean ride-sharing platform, Grab. On this platform, users request and pay for rides through a mobile phone application that mediates their transactions with service provider. In this context, the service providers are the drivers offering transportation services. Signaling theory (Spence 1973) serves as a framework to understand how star ratings, a kind of reputation cue, affect users' willingness to use service providers. The current study addresses the broad question, what qualities of service providers do reputation cues signal? Focal to this study, do user perceptions of those qualities explain

why reputation cues affect their willingness to use service providers? Also, do these effects depend on how much users pay attention to reputation cues? Toward answering these questions, the literature review draws on prior research showing reputation cues affect consumer beliefs (Akdeniz et al. 2013; Adomavicius et al. 2018) and beliefs affect behavioral intentions (Sullivan and Kim 2018). We draw those lines of research together, arguing signals affect decisions to engage in collaborative consumption through influencing specific beliefs about service providers. Finally, we argue those effects depend on users paying attention to reputation cues.

### 1.1. The Sharing Economy

Sharing is an everyday activity involving the collective use of a commodity. In some instances of sharing, groups of individuals cooperate to acquire a commodity whose use they apportion into shares (Guillén et al. 2012; L'Aoustet and Griffet 2004). In other instances of sharing, individuals possess commodities they make available for collective use. This often involves commodities whose sharing does not overly deprive the owners of their usefulness or value. Some examples of these commodities include private vehicles, accommodations, appliances, and physical labor. The process of distributing such commodities is at the heart of the sharing economy, which has existed since ancient times and has received growing attention in recent literature (Cheng 2016; Frenken and Schor 2017).

Sundararajan (2016) articulated five characteristics of the sharing economy. First, it creates markets where individuals exchange goods and services for other things of value. Second, it maximizes capital utilization, where goods and services are used at or near their capacity. Third, it is decentralized and depends on crowd-based networks. Fourth and fifth, it blurs the distinctions between personal possession and commercial object and between traditional and non-traditional employment. He acknowledged his definition is capitalism-centric and does not address sharing behavior in the traditional sense of the word. The apparent conflict between commercial activities and sharing behavior forms the nexus of articulated five characteristics Lessig (2008) calls the hybrid economy, in which the internet increasingly facilitates the commercialization of sharing. In a similar line of scholarship, Bardhi and Eckhardt (2012) distinguished between sharing and access-based consumption. Whereas neither activity involves personal ownership of a commodity, sharing entails joint ownership and shared responsibility over the commodity. On the other hand, access involves the owner of a commodity providing its temporary use to others through a marketplace. The general forms of the hybrid economy and access-based consumption comport with Sundararajan's (2016) characterization of the sharing economy.

Motivations for individuals to participate in the sharing economy may include economic, social, and environmental factors (Schor and Fitzmaurice 2015; Hamari et al. 2016; Sung et al. 2018). The economic motivation has a lot to do with efficiency that comes from reducing or eliminating middlemen in exchanges. The social motivation is related to social networking and building social capital. Finally, individuals may be motivated to share because it can reduce environmental burdens of consumption. Böcker and Meelen (2017) studied the factors motivating individuals to engage in different kinds of sharing, both as providers and users. They found motivations to both provide and use accommodation sharing primarily reflected economic considerations, but also had a social aspect. On the other hand, whereas providing ride sharing was primarily motivated by environmental and social factors, the use of ride sharing was relatively balanced among the three motivations.

Participating in the sharing economy is distinct from participating in many other kinds of sharing. This is in part because shared service providers are often strangers to the users. Schor (2016) referred to this phenomenon as "stranger sharing," which she differentiated from traditional modes of sharing that occur among individuals within social networks. If there is any novel aspect of the sharing economy, it is not related to the fact people exchange commodities, but that it has become easy for people to exchange commodities with strangers (Frenken and Schor 2017). Because service providers and users do not know each other, the act of sharing entails some risk, which can demotivate participation in the sharing economy. These risks can include one party failing to make their end of the agreed upon exchange and more severe risks, like unsafe driving and physical violence (e.g., Dills and Mulholland 2018). From

a user perspective, service quality and trustworthiness can enhance satisfaction with the service (Möhlmann 2015), and having that information about a service provider can alleviate some of the perceived risk associated with stranger sharing (Ert et al. 2016). Many shared services and other online platforms have addressed this by giving users the means to share and access ratings and other feedback about service providers (Schor 2016; Jøsang et al. 2007). Rating systems have given many users the confidence to pursue exchanges with strangers (Frenken and Schor 2017). This is because ratings and feedback function as signals about positive or negative attributes of service providers. Service users can make judgments based on these signals, affecting their willingness to engage with individual service providers. Signaling theory provides a framework to explain this process.

### 1.2. Signaling Theory

In the sharing economy, users are at an information disadvantage relative to providers about the services being offered. This proposition applies to buyers and sellers in general. Spence (1973) distinguished between two kinds of information influencing buyers, indices and signals. Indices are unalterable attributes of a good or service, typically objective features defining the type of product, while signals are observable attributes of a good or service the seller can manipulate to distinguish it in the marketplace. Product durability is an example of a signal because it is something producers can enhance, hoping the gains in product reputation and sales will offset the cost of making it more durable.

Spence (1973) discussed this cost-benefit balance in terms of signaling cost, which refers to the resources—psychological, physical, monetary, temporal, etc.—involved in creating signals. Costly signals are reliable indicators of seller quality because only higher-quality sellers can afford them. In the context of the ride sharing, signal costs might include things like maintaining a clean car and having a friendly demeanor, which would involve temporal and emotional resources. Since many activities in the sharing economy involve users making decisions about strangers, they are often unable to directly assess potential costly signals in advance. Rather, they rely on things like reputation cues, which may imply the presence of costly signals in service providers. Thus, signaling theory is also useful for explaining how reputation cues act as signals to cue specific beliefs about service providers (Slee 2017; Teubner et al. 2017).

### 1.3. Reputation Cues

Many online marketplaces employ reputation systems signaling to buyers the qualities of products and sellers (Drover et al. 2018; Jøsang et al. 2007). Although these informational cues are more limited than those available by meeting sellers in person (Jøsang et al. 2007), they can provide buyers with valuable information about products and services. The upshot is reputation cues affect sales (Ye et al. 2009; Chevalier and Mayzlin 2006). Consistent with the tenets of signaling theory, Wu et al. (2013) found product reviews increase risk-averse buyers' willingness to purchase by reducing their uncertainties about the product and seller. The current study partially replicates this effect in the context of a ride-sharing service, focusing on the relationship between the reputation of the service provider (i.e., the driver) and willingness to use their service (i.e., purchase a ride). We expect to see shared service users will have a greater willingness to use a service provider with a high reputation score than one with a low reputation score. This is obvious to predict because reputation affects beliefs about a service provider. But what kinds of beliefs do they affect? The current study looks at perceived service quality, which can affect buyer intention in the context of the sharing economy (Ert et al. 2016; Xie and Mao 2017; Zeithaml 2000).

### 1.4. Perceived Service Quality

Buyers form expectations about sellers, which may reflect what service they receive and how they receive it. Grönroos (1984) described these two kinds of expectations as the technical and functional qualities of a service. Together these qualities affect perceived service quality, which arises at the intersection of what buyers expect to receive and what they perceive to have received.

Parasuraman et al. (1988) built on this conceptual foundation and explicated five dimensions of service quality: tangibles, reliability, assurance, responsiveness, and empathy. Respectively, these dimensions reflect the condition of physical facilities, accuracy and dependability, ability to inspire trust and confidence, willingness to assist customers, and the care and attention given to each customer. Despite some early criticisms (Buttle 1996), this SERVQUAL model has become a widely used framework to describe and measure service quality (Carrillat et al. 2007).

Sellers often try to distinguish themselves in the marketplace by adjusting what they sell and how they engage with buyers (Parasuraman et al. 1988). In this respect, perceived service quality functions as a signal, which sellers and the marketplace develop. Truong et al. (2017) found the perceived quality of food, atmosphere, cleanliness, and employees affected the perceived service quality of a restaurant. Ho and Wei (2016) found authentic experiences with sellers affect perceived service quality, which they described as a function of signal credibility. With stranger sharing the process is similar, but the buyer must learn about service quality indirectly. There is some evidence ratings and reviews affect perceived service quality (Browning et al. 2013). Thus, we also expect to see shared service users will perceive greater service quality of a service provider with a high reputation score than one with a low reputation score.

### 1.5. Mediation of Reputation

The effect of perceived service quality is intuitive: buyers generally prefer to engage with sellers whom they expect to have a high standard of service. Researchers have shown in many contexts the relationship between perceived service quality and willingness to use a service. Recent examples of these contexts include air travel (Hussain et al. 2015), government e-services (Sharma 2015), restaurants (Truong et al. 2017), and service innovations (Storey et al. 2016). The current study is interested in these relationships not as an attempt to replicate those prior studies, but rather to link the expected effects we noted earlier. We assume willingness to use a service provider and perceived service quality are related to each other and both are affected by reputation scores. If those assumptions are correct, then perceived service quality may function as a pathway by which reputation cues affect user willingness. We suggest a good reputation leads to greater willingness partly because of its effects on perceived service quality. Prior research has not examined this mediation process.

**Hypothesis 1 (H1).** *The effect of reputation score on willingness to use the service provider is mediated by perceived service quality.*

### 1.6. Attention to Reputation Cues

We have argued reputation cues are signals sellers can manipulate to distinguish themselves in the marketplace. Signal effectiveness depends partly on how sellers communicate with buyers. It also depends on what buyers do with that information (Connelly et al. 2011).

Marketing research has described a hierarchy of effects concerning the cognitive, affective, and motivational processes that lead to purchasing behavior (Lavidge and Steiner 1961; McGuire 1976; Engel et al. 1995). Although more than a dozen models describe this process, most identify attention as a starting point of information processing (Wijaya 2015). The idea is straightforward: individuals need to pay attention to information before they can interpret and act on it.

The hierarchy of effects can explain the process of signaling from the perspective of buyers, who must pay attention to the signal information prior to forming beliefs and purchase intentions. There is some evidence supporting this assertion. Gulati and Higgins (2003) developed a theory arguing signals from firms about their investment value have the greatest impact when investors are paying careful attention to them. Drover et al. (2018) developed a similar theory to explain how individuals process signals from organizations. They based their theory on dual process models, arguing informational needs affect the amount of attention individuals pay to signals, which affects confidence to make a decision. Chattopadhyay and Nedungadi (1992) found experimentally that participants who paid

attention to an advertisement had a more positive brand attitude. Common among these works, attention is a precursor to individuals acting on signal information. We apply this notion to explain how reputation cues affect beliefs and behavioral intention in the sharing economy.

**Hypothesis 2 (H2).** *The more individuals pay attention to reputation cues, the larger the effect of reputation score on (a) perceived service quality and (b) willingness to use the service provider.*

### 1.7. Study Context

This study takes place in the context of the Singaporean ride-sharing company, Grab. The company was founded as MyTeksi in Kuala Lumpur in 2012 (Wee 2017). In 2014, Grab moved its headquarters to Singapore (Kaushik 2020). At the time, Grab's main competitor was Uber, whose operations in Southeast Asia it acquired in a 2018 merger (Grab 2018). Although other competitors remain, as of mid-2018, Grab had roughly a 60% market share in Southeast Asia (Huang 2018).

## 2. Materials and Methods

### 2.1. Power Analysis

The current experiment has two repeated measures and a continuous between-subjects moderator. We used G*Power version 3.1, written by Franz Faul of the University of Kiel, Germany, to estimate the minimum sample size to detect a moderate effect ($f = 0.25$), allowing 5% Type I and Type II error probabilities and a conservative assumption of uncorrelated repeated measures. Because G*Power does not have an option to include continuous moderators in repeated measures models, we treated the continuous moderator as a two-level between-subjects factor. The recommended sample size for repeated measures and a within-between interaction is 106.

### 2.2. Sample

The study recruited participants through convenience sampling from an undergraduate research panel at a large research university in Singapore and from students' personal networks in Singapore. This resulted in a sample of 221 respondents, which is overpowered given our sample size analysis and large enough to detect effects as small as $f = 0.17$. The participants were mostly female (74%) and ranged in age from 18 to 62 years (Mdn. $= 21$, M $= 22.81$, SD $= 6.78$).

### 2.3. Procedure

Prior to the experiment, we conducted a pretest, which validated our focus on perceived service quality (see Appendix A). We used an online Qualtrics survey to conduct the experiment. Participants viewed in random order the profiles of six different drivers. Each driver had a different low or high star rating. After viewing each profile, participants answered questions about perceived service quality and willingness to use the service provider. They also answered a manipulation check to evaluate whether they perceived the high and low ratings accordingly.

### 2.4. Manipulations

We created mockups of driver profiles, replicating the design of those appearing on the Singaporean ride sharing service, Grab. The profiles included the driver's name, picture, and average star rating. The drivers with low ratings had 3.1, 3.2, or 3.3 stars. The drivers with high ratings had 4.8, 4.9, or 5.0 stars. We based the range of ratings on feedback in the pretest that most users give drivers four- or five-star ratings. In the real world, Grab would most likely suspend drivers with ratings near three stars. However, for the purposes of testing our hypotheses, such a rating would be unambiguously low and establish the direction of the proposed mechanism, which is an acceptable use of a laboratory experiment even when external validity is low (Kessler and Vesterlund 2015, p. 394). We rotated the driver names and profile pictures using a Latin square design. This removed potential confounding

effects related to driver name or appearance. The manipulation check was a single Likert item: "This driver has a good rating." Response options ranged from 1 (strongly disagree) to 7 (strongly agree). Agreement with that item was higher for service providers with high reputation scores (M = 6.56, SD = 0.48) than low reputation scores (M = 3.91, SE = 1.27). The difference in means was significant (ΔM = 2.64, SD = 1.36, 95% CI [2.47, 2.83]), $t$ (220) = 29.00, $p$ < 0.001, suggesting the manipulation was successful.

### 2.5. Pre-Score Measures

We measured attention to reputation scores using a single item: "When purchasing goods and services online, how much attention do you pay to ratings?" Response options ranged from 1 (none at all) to 5 (a great deal; M = 3.72, SD = 0.87). This item was measured only one time prior to participants viewing the stimuli.

Although the manipulations focused on service providers, it was possible that prior experience with the ride sharing platform influenced the dependent variables. Therefore, we measured two control variables. The first variable simply asked respondents, "How often do you use ride-share services? (e.g., Grab, Uber, etc.)?" This measured the frequency of past use. Response options ranged from 1 (never) to 5 (all of the time; M = 3.01, SD = 0.88). The second variable was a composite measure of attitude toward the platform and included four items: "Grab is sincere," "( … ) honest," "( … ) concerned with the interests of its customers," and "( … ) receptive to the needs of its users." Response options ranged from 1 (strongly disagree) to 7 (strongly agree; M = 4.97, SD = 0.98).

### 2.6. Repeated Measures

We measured perceived service quality with 12 Likert items adapted from prior research (Zhang et al. 2013; Daudi et al. 2018). The items measured participants' beliefs the driver "has a well-maintained car," "has a comfortable car," "gets me to my destination quickly," "arrives on time," "is polite," "is friendly," "is quick to provide help," "maintains professional boundaries," "is attentive to my needs," "drives safely," "is trustworthy," and "is competent." Response options ranged from 1 (strongly disagree) to 7 (strongly agree). See Table 1 for item wording and descriptive statistics.

We adapted an existing measure of willingness to use the service provider (Möhlmann 2015), which we measured with three Likert items: "I would want a ride from this driver," "I would accept a ride from this driver," and "If I got this driver, I would cancel" (reverse-coded). Response options ranged from 1 (strongly disagree) to 7 (strongly agree). See Table 1 for item wording and descriptive statistics of the composite measures.

### 2.7. Measurement Reliability

We conducted factor analysis of the 19 measurement items, using maximum likelihood estimation and direct oblimin rotation. The items measuring perceived service quality loaded on a first factor (AVE = 0.79), which had an eigenvalue of 10.78 and explained 57% of the variance in the full set of items. Those items had excellent reliability (Cronbach's $\alpha$ = 0.98). The three items measuring willingness to use the service provider loaded on a second factor (AVE = 0.72), which had an eigenvalue of 2.78 and explained an additional 15% of the variance. Those items had good reliability (Cronbach's $\alpha$ = 0.89). The four items measuring attitude toward the platform loaded on a third factor (AVE = 0.72), which had an eigenvalue of 1.72 and explained an additional 9% of the variance. Those items had good reliability (Cronbach's $\alpha$ = 0.87). Most factor cross-loadings were close to zero. The first two factors were positively correlated ($r$ = 0.52). The third factor had small negative correlations with the first factor ($r$ = −0.10) and the second factor ($r$ = −0.17). See Table 1 for item factor loadings.

**Table 1.** Measurement items, descriptive statistics, and factor loadings.

| Scale/Wording | M (SD) Overall | M (SD) Low Rating | M (SD) High Rating | $\lambda_1$ | $\lambda_2$ | $\lambda_3$ |
|---|---|---|---|---|---|---|
| Perceived service quality | | | | | | |
| has a well-maintained car | 4.88 (0.84) | 4.30 (0.91) | 5.46 (1.07) | 0.81 | 0.02 | −0.05 |
| has a comfortable car | 4.90 (0.83) | 4.35 (0.91) | 5.46 (1.07) | 0.81 | 0.02 | −0.03 |
| gets me to my destination quickly | 4.89 (0.81) | 4.28 (1.00) | 5.51 (1.03) | 0.89 | −0.02 | −0.06 |
| arrives on time | 4.98 (0.75) | 4.31 (1.00) | 5.66 (0.94) | 0.88 | 0.01 | −0.01 |
| is polite | 5.06 (0.75) | 4.39 (1.01) | 5.72 (0.88) | 0.92 | 0.03 | 0.04 |
| is friendly | 5.07 (0.76) | 4.42 (1.03) | 5.71 (0.89) | 0.87 | 0.06 | 0.01 |
| is quick to provide help | 4.86 (0.79) | 4.25 (0.93) | 5.48 (1.00) | 0.96 | −0.04 | 0.06 |
| maintains professional boundaries | 4.93 (0.77) | 4.37 (0.90) | 5.49 (0.97) | 0.91 | 0.00 | 0.02 |
| is attentive to my needs | 4.80 (0.78) | 4.23 (0.90) | 5.38 (1.02) | 0.96 | −0.02 | 0.07 |
| drives safely | 5.03 (0.75) | 4.37 (1.02) | 5.70 (0.90) | 0.89 | 0.00 | −0.01 |
| is trustworthy | 4.98 (0.77) | 4.33 (1.00) | 5.62 (0.94) | 0.91 | 0.00 | 0.01 |
| is competent | 5.07 (0.76) | 4.41 (0.98) | 5.72 (0.93) | 0.86 | 0.05 | −0.05 |
| Willingness to use the service provider | | | | | | |
| I would want a ride from this driver | 5.34 (0.72) | 4.52 (1.22) | 6.16 (0.63) | 0.22 | 0.73 | −0.01 |
| I would accept a ride from this driver | 5.63 (0.73) | 4.98 (1.22) | 6.28 (0.60) | 0.02 | 0.98 | −0.05 |
| If I got this driver, I would cancel | 5.66 (0.96) | 5.11 (1.42) | 6.21 (0.90) | −0.06 | 0.82 | 0.03 |
| Attitude toward the platform | | | | | | |
| sincere | 5.05 (1.11) | | | 0.10 | −0.06 | 0.90 |
| honest | 5.07 (1.20) | | | 0.07 | −0.03 | 0.83 |
| concerned with the interests of its customers | 4.74 (1.17) | | | −0.02 | −0.01 | 0.73 |
| receptive to the needs of its users | 5.04 (1.18) | | | −0.13 | 0.09 | 0.66 |

Note. $\lambda_k$ = factor loading on $k$ factor. Items measuring perceived service quality had the common stem, "Based on the driver profile, the driver:" Items measuring attitude toward the platform had the common stem, "Grab is:" The factor analysis of repeated measures was based on the pooled measures (i.e., the "overall" measures).

## 3. Results

We used SPSS Statistics version 25, published by IBM Corp. in Armonk, New York, NY, USA, to conduct all our analyses. First, we examined if the treatment effects depended on either of the control variables. We conducted repeated measures analysis of covariance (ANCOVA) to compare scores on perceived service quality (analysis 1) and willingness to use the service provider (analysis 2) when service providers has low versus high reputation scores. Results of the first analysis showed the treatment effect on perceived service quality was unrelated to the self-reported frequency of ride share use, $F (1, 218) = 0.14$, $p = 0.71$, and unrelated to attitude toward the platform, $F (1, 218) = 0.25$, $p = 0.62$. Results of the second analysis showed the treatment effect on willingness to use the service provider was unrelated to the self-reported frequency of ride share use, $F (1, 218) = 0.05$, $p = 0.82$, and unrelated to attitude toward the platform, $F (1, 218) = 0.09$, $p = 0.77$.

We expected that willingness to use a service provider and perceived service quality are higher when service providers have high versus low reputation scores. We formally tested these effects to link our study with prior research. We used paired $t$-tests, which is an appropriate analysis when comparing means on two repeated measures. Consistent with our first expectation, there was a greater willingness to use service providers with high reputation scores (M = 6.22, SD = 0.61) than low reputation scores (M = 4.87, SD = 1.21) The difference in means was significant ($\Delta M = 1.35$, SD = 1.22, 95% CI [1.19, 1.51]), $t (220) = 16.32$, $p < 0.001$. Consistent with our second expectation, there was greater perceived service quality of service providers with high reputation scores (M = 5.58, SD = 0.87) than low reputation scores (M = 4.33, SD = 0.87) The difference in means was significant ($\Delta M = 1.24$, SD = 1.01, 95% CI [1.11, 1.38]), $t (220) = 18.36$, $p < 0.001$.

We used the mediation and moderation for repeated measures (MEMORE) macro for SPSS (Montoya and Hayes 2017) to test H1 and H2. This macro simplifies the analysis of repeated mediators and continuous moderators in repeated measures models. The results include 95% confidence intervals using 5000 bias-corrected bootstrap samples. The test of H1 used MEMORE model 1 (mediation)

involved decomposing the total effect of reputation score on willingness to use the service provider. After controlling for perceived service quality, the direct effect was smaller but remained significant ($\Delta M = 0.24$, SD = 1.23, 95% CI [0.07, 0.40]), $t$ (218) = 2.74, $p = 0.007$. In support of H1, the indirect effect of reputation score was also significant ($\Delta M = 1.11$, SD = 1.41, 95% CI [0.92, 1.530]). The indirect effect was 83% of the total effect, suggesting a high degree of mediation. Figure 1 shows the effects decomposition.

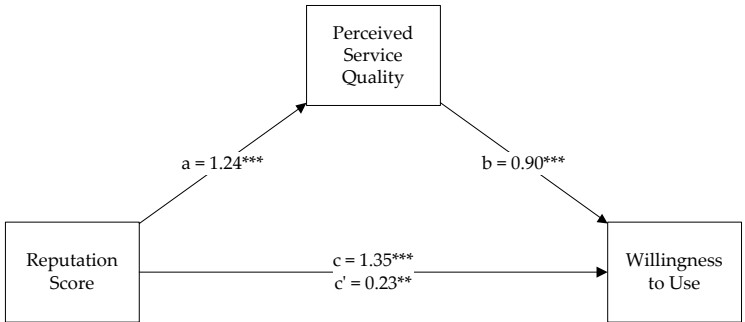

**Figure 1.** Mediation model. ** $p < 0.01$. *** $p < 0.001$.

The test of H2 used MEMORE model 2 (moderation). In support of H2 (a), the more participants reported paying attention to ratings, the stronger the effect of reputation score on perceived service quality (B = 0.17, SE = 0.08, 95% CI [0.01, 0.32], $p = 0.033$, $\eta^2_p = 0.02$). This small moderation effect is apparent when comparing the effect of reputation score on perceived service quality at high self-reported attention and low self-reported attention (see Figure 2). The difference in perceived service quality between low- and high-rated drivers is the largest when participants reported they pay a great deal of attention to ratings. That difference became smaller when participants reported paying no attention at all to ratings. Failing to support H2 (b), the effect of reputation score on willingness to use the service provider was unrelated to self-reported attention to ratings (B = 0.14, SE = 0.10, 95% CI [−0.05, 0.32], $p = 0.16$).

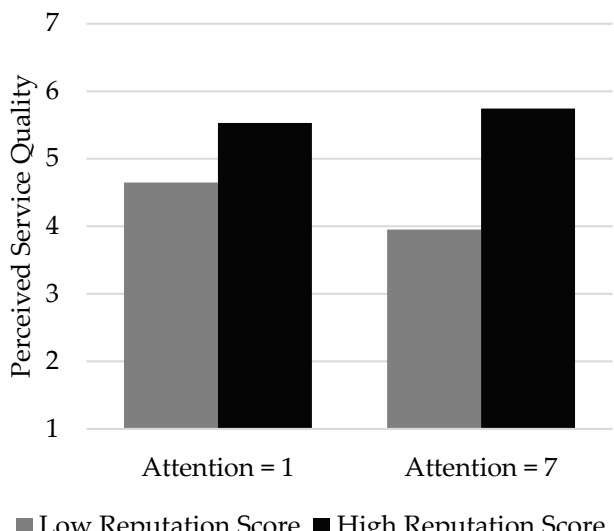

**Figure 2.** Effects of reputation score at the minimum and maximum attention scores.

## 4. Discussion

This study investigated the signaling effects of reputation cues in the sharing economy, focusing on the context of the Singaporean ride sharing platform, Grab. It considered direct, mediated, and moderated pathways affecting perceived service quality and willingness to use a service provider. Analysis of experimental data largely supported our predictions, showing reputation cues affect

perceive service quality, which affects willingness to use a service provider. Furthermore, the effect of reputation cues on perceived service quality is stronger among participants who say they pay more attention to driver ratings. These findings have implications for signaling theory, particularly how reputation cues may imply the presence of costly signals in service providers. There are additional implications for dual process models of information processing. Some practical implications concern the utility of reputation systems for helping the users of shared services make informed decisions and the service providers enhance their marketability.

### 4.1. Interpreting the Present Findings

Our first analyses concerned the direct effects of reputation cues on willingness to use the service provider and perceived service quality. This portion of our study replicated prior work and found, consistent with those prior studies, reputation cues signal qualities of a service provider. Those signals affect users' beliefs about the service provider—in this case, beliefs regarding service quality—and willingness to use the service from that provider (Ert et al. 2016; Xie and Mao 2017; Zeithaml 2000). Although prior research has shown similar effects in many commerce-related contexts, including e-commerce and the sharing economy, they are worth replicating in the context of ride sharing. This is because the utility of reputation cues as signals and the nature of reputation can vary among contexts. For example, in the context of accommodation sharing, assurance is related to concerns about last-minute cancellations, whether keys to the accommodation will be provided as agreed, and whether the accommodation is legitimate (Phua 2018). In the context of ride sharing, the safety aspect of assurance seems important. There are likely other dimensions of perceived service quality that may be unique to this context, which future research could tease out.

The more novel contributions of our study came in testing our two hypotheses, which concerned mediated and moderated effects of reputation cues. The first hypothesis stated perceived service quality mediates the effect of reputation cues on willingness to use the service provider. Prior research has shown each direct path within this mediation model—reputation cues affect perceived service quality (Browning et al. 2013; Sparks and Browning 2011), which in turn is related to willingness to use (Sharma 2015; Bélanger and Carter 2008)—but has not examined the indirect paths that link them. The current analysis showed perceive service quality significantly explained the effect of reputation cues on willingness to use the service provider. This finding clarifies the mental processes shared service users employ when evaluating the signaled attributes of service providers. In particular, the overall appraisal of service providers reflects beliefs about their attributes that may imply the presence of costly signals. Relevant to the current study, building service quality consumes resources such as time and effort (Li et al. 2015). Individuals are more trusted when they engage in costly signaling (Hall et al. 2015), especially when those signals indicate prosocial intent (Bliege Bird and Power 2015). Thus, a good reputation may convey costly signals, which users of shared services can infer from reputation cues. Those inferences affect beliefs about service providers, which subsequently affect user willingness. This would provide one explanation of why reputation cues affect willingness.

Our second hypothesis stated the effects of reputation cues depend on how much the shared service users pay attention to them. This prediction comports with the hierarchy of effects model (Wijaya 2015), which states the effects of messages depend on audiences being exposed to them and paying attention. The current findings supported this perspective, showing the effects of reputation cues were stronger among respondents who said they pay more attention to such cues. This suggests the effects of reputation cues occur through systematic information processing. Yet, the idea of an information *cue* implies certain heuristic value. This has implications for dual process models of thought. For example, the heuristic-systematic model of information processing (Chaiken 1980) states people tend to superficially process information to make quick judgments, relying on heuristic processing. But when individuals have important judgments to make, they may engage more effort in evaluating relevant information. This involves systematic processing. In the context of ride sharing, many users likely process star ratings more heuristically, making snap judgments about the qualities of

service providers. Other users may have concerns about service quality or other aspects of a service and be motivated to carefully consider their decision. Those users ought to pay more attention to star ratings and process them more systematically, drawing inferences about specific attributes of service providers before deciding whether to use their services. The current study is unable to test these assertions, and this could be an interesting topic of future studies.

Another perspective on attention suggests a different causal ordering. Rather than attention causing information effects—which include things like attitude change—attitude causes attention. This is consistent with the cognitive mediation model (McLeod et al. 1994), which suggests audiences respond to information based on prior orientations, which can include values, beliefs, and attitudes. Vaughan et al. (2016) found brand users had better recall than non-users of brand advertisements, suggesting prior orientations to the brand influenced their attention. Does that perspective help explain the current findings? To an extent, yes. According to the reinforcing spirals framework (Slater 2007), selective information seeking affects and is affected by related attitudes and behaviors. This means people pay attention to information because they are already oriented to it and their attention affects subsequent orientations. Yet, the current findings seem most aligned with the latter portion of that framework. We cannot argue the experimental manipulations caused attention. This is because we measured attention prior to the manipulation. What our results showed is the treatment effects were stronger among those who said they pay more attention to ratings. But in the real world, it is likely the case that people are influenced more by information when they are paying attention and the extent of that influence affects their subsequent attention.

### 4.2. Broader Implications

Whereas this study focused on how shared service users form beliefs and make decisions based on reputation cues, it would be informative if future research focused on shared service providers, answering two questions: (1) How hard do providers work to cultivate a good reputation and (2) what drives them to do so? Intuitive answers are they work as hard as they need to maintain an acceptable rating, and they do so to achieve optimal bookings. Is their effort simply tit-for-tat, where they expect to receive a good review after providing a good service? This explanation aligns somewhat with the notion of reciprocal altruism, in which individuals share resources with a community, expecting they will receive equivalent benefit from it in the future (Bird and Smith 2005).

An alternative explanation draws on costly signaling theory. In the context of the sharing economy, this would mean service providers improve their service because it affects their social standing. Yet, not all service providers can afford costly signals, and the equilibrium strategy for low-quality service providers is to avoid costly signaling simply because they cannot afford it (Li et al. 2015; Spence 1973). The choice of whether to signal or not ought to affect the ratings users give service providers, and low-quality service providers adopting the equilibrium strategy will fail to cultivate a good reputation. This implies somewhat of a downward spiral, which the current findings do not address directly. Yet, it would explain why in 2019, the average rating of Grab drivers was around 4.8 (Grab 2019). When costly signals are all service providers can use to differentiate themselves in the marketplace, only the strong will survive.

## 5. Limitations and Conclusions

This study showed reputation cues function as signals about qualities of ride-sharing service providers. In this case, star ratings affect beliefs about service quality, which may explain willingness to use a service provider. Of course, there may be other qualities that come to mind when ride-sharing service users evaluate star ratings. For example, does a high reputation score suggest the driver is authentic? Bucher et al. (2018) found users of Airbnb who had authentic experiences were less likely to be influenced by negative aspects of their stays, such as unclean or cluttered accommodations. That study was not interested in effects of reputation cues, but it is possible service users infer levels of authenticity from reputation scores. Addressing such other qualities of service providers would be a

good avenue for future research. Another limitation is the current moderation model was simplistic. There are many potential moderators other than attention, such as trust in the service platform and trust in other users to give unbiased ratings. The current findings make the most sense when users trust the reputation scores are accurate. What happens when users distrust the scores or platform? Two other possible moderating factors are time pressure and information needs, where service users may react differently to reputation scores when they have less time to process them or have more need for information. Future research can address these moderators. Despite their limitations, the current findings are compatible with prior research on the sharing economy and the tenets of signaling theory. They show reputation cues help service users form actionable beliefs about service providers, especially when they pay attention to those cues. Beyond the theoretical implications, these findings suggest reputation cues are essential components of shared service platforms and service providers can efficiently maintain a good rating by understanding what factors users associate with good service.

**Author Contributions:** Conceptualization, S.R., J.Y.C.T., T.F.P.; methodology, J.Y.C.T., T.F.P.; formal analysis, S.R.; writing—original draft preparation, S.R., J.Y.C.T., T.F.P.; writing—review and editing, S.R.; visualization, S.R. All authors have read and agreed to the published version of the manuscript.

**Funding:** This research was funded by an internal grant by the Wee Kim Wee School of Communication and Information at Nanyang Technological University, Singapore.

**Conflicts of Interest:** The authors declare no conflict of interest. The funders had no role in the design of the study; in the collection, analyses, or interpretation of data; in the writing of the manuscript, or in the decision to publish the results.

## Appendix A

We conducted a pretest to gauge the salience of service quality to users of a ride-sharing platform. This helped ensure exhaustive operationalizations of these concepts, which reputation cues may signal. The pretest involved brief face-to-face interviews with 51 ride-share users intercepted at a downtown Singapore shopping mall upon their exit from a shared vehicle. We asked each respondent how many stars they would rate their last ride and then briefly probed about their reasons for giving that rating.

We identified three themes aligning with the SERVQUAL model. Many respondents emphasized efficiency, timeliness, road-familiarity, and a good sense of direction. These driver attributes are related to reliability. Another common theme was about assurance and responsiveness, with respondents emphasizing the detriment of having an impolite driver. Less common were responses related to tangibles and empathy, with only a few respondents noting things like vehicle cleanliness and personalized service as contributing to their ride experience. A fourth and dominant theme was related to assurance. Many respondents took note of the driver's behavior and would give a lower rating for dangerous actions such as speeding, tailgating, and using their phone while driving. It is reasonable to infer users would be less willing to use a service provider they cannot trust to ensure their safety. There were some additional idiosyncratic responses. For example, one respondent mentioned giving a five-star rating so as not to affect the driver's bonus. However, such responses may be specific to the remuneration package of the company under study and that finding will not necessarily generalize to other platforms in the sharing economy.

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
