# Peer review of "Reputation Cues as Signals in the Sharing Economy"

_socsci, doi:10.3390/socsci9040049_

Round 1
Reviewer 1 Report
Dear authors,
I appreciate the opportunity to review this interesting manuscript which is well written. The research design is appropriate, the methods adequately described, the results clearly presented and the conclusion supported by the results. To strengthened the paper please pay attention to followings:
- the article should be reordered according to the Instructions for Authors (Introduction, Materials and Methods, Results, Discussion, Conclusion and References)
- references should be numbered in order of appearance in the text (including citations in tables and legends) and listed individually at the end of the manuscript. In the text, reference numbers should be placed in square brackets [ ], and placed before the punctuation; for example [1], [1–3] or [1,3]. For embedded citations in the text with pagination, use both parentheses and brackets to indicate the reference number and page numbers; for example, [5] (p. 10), or [6] (pp. 101–105).
Author Response
Thank you for your review. Please see the attached document for my response. Note that I retained the Author-Date format of my citations based on the instructions to authors for this journal. Other MDPI journals use different formats.

Reviewer 2 Report
The authors presents an investigation about the sharing economy. This topic is very interesting and is one of the lines of research that need more scientific contributions. Also, the analysis provides some conclusions that may be of greater interest to readers of this journal.
However, The methodology does not seem the most appropriate. In this regard, I recommend completing this research with additional methodologies. For example, complete your work with a literature review, as for example you can read in these works:
Cheng, M. (2016). Sharing economy: A review and agenda for future research. International Journal of Hospitality Management, 57, 60-70.
Palos-Sanchez, P.R. & Correia, M.B. (2018). The collaborative economy based analysis of demand: study of Airbnb case in Spain and Portugal. Journal of Theoretical and Applied Electronic Commerce Research 13 (3), doi: 0.4067 / S0718-18762018000300105
J. Barnes and J. Mattsson, Understanding current and future issues in collaborative consumption: A four-stage Delphi study, Technological Forecasting and Social Change, vol. 104, pp. 200-211, 2016.
Regarding the methodology, I do not understand why the authors do not follow the logical order and present the methodology before the analysis of the results.
The authors do not present the type of sampling applied, nor do they present how bias has been eliminated. What type of research analysis are they conducting?
I recommend calculating the sample size using the gpower tool (central and non-central distributions-F-test).
Read on:
F. Faul, E. Erdfelder, A. Buchner, and A.-G. Lang, “Statistical
power analyses using G*Power 3.1: tests for correlation and
regression analyses,” Behavior Research Methods, vol. 41, no. 4,
pp. 1149–1160, 2009.
Reyes-Menendez, A., Palos-Sanchez, P. R., Saura, J. R., & Martin-Velicia, F. (2018). Understanding the influence of wireless communications and Wi-Fi access on customer loyalty: A behavioral model system. Wireless Communications and Mobile Computing, 2018. (Page 5)
This way they can calculate the power for mediation.
Nor do they justify why they choose this type of statistical analysis and not another of the multivariate type.
I congratulate the researchers and I wish them luck.
Author Response
Thank you for your review. Please see the attached document for my response.

Reviewer 3 Report
This is a well-written paper that has been reasonably well-conducted. However, I feel that it does not really contribute very much to what we already know. It has two major omissions, in my view, at present. Firstly it does not justify why its first two RQs are necessary, why what has been established about influences on sharing behaviour may not apply to ride sharing in Singapore. Secondly there are many covariates present in their study that are not acknowledged: I feel that at the very least a Limitations section needs to be in the paper.
Author Response

(The authors gave the same response as above.)

Round 2
Reviewer 2 Report
I sincerely believe that authors should improve the literature review and strengthen the concepts on sharing economy. I understand perfectly that you want to be concise, but it is very appropriate for the reader's understanding that you improve and expand this section.
The real value of this work lies in the fact that it investigates this new form of economy, which is currently being developed in all countries of the world.
Author Response
Thank you for the quick review turnaround. The original submission you reviewed dedicated only 308 words to the section on the sharing economy. This version has more than doubled the length of that section.
Reviewer 3 Report
I appreciate that the authors have acknowledged and somewhat addressed some of the concerns and limitations I felt were present in the paper.
They suggest that one possible additional influence on willingness to pay is intra or extra- version but it is not clear whether this is of the service provider or the potential user of the service. I think that this needs to be clarified. Also, why was this particular dimension of Personality mentioned? There are other personality factors that could be influential.
I still feel that other information that I suggested should be given about the context of ride-sharing in Singapore should be added, like is ride-sharing, at the time of the study, still relatively novel (when reputation would still be being formed) or well-established? A limitation of the methodology was that whether the respondents were experienced users of ride sharing or not, was not measured. If they are experienced, they may have already formed views about the reputation of the booking service provider (e.g. Uber) for whom the service provider (driver) is associated and that this influences willingness to use the service. You suggest that those who pay attention to advertisements consequently have a favourable attitude to the advertiser - but the causality is the other way round, i.e. those with a favourable attitude to a brand, due to their being a user of it, notice its advertising more than non-users is well-established ( e.g. Vaughan, K., Beal, V., and Romaniuk, J. 2016. “Can Brand Users Really Remember Advertising More Than Nonusers? Testing an Empirical Generalization across Six Advertising Awareness Measures.” Journal of Advertising Research 56 (3):311-20 )
There are two corrections/edits to be made. In Discussion, line 2, what is the word "Grab" - should it be there?? and in 1.6, line 1, the word 'them" needs to be deleted.
Author Response
Thank you for your quick review! Please see the uploaded response letter.
